# GANet:Glyph-Attention Network for Few-Shot Font Generation

## Abstract

Font generation is a valuable but challenging task, it is time consuming and costly to design font libraries which cover all glyphs with various styles. The time and cost of such task will be greatly reduced if the complete font library can be generated from only a few custom samples. Inspired by font characteristics and global and local attention mechanismWang et al. (2018), we propose a glyph-attention network (GANet) to tackle this problem. Firstly, a content encoder and a style encoder are trained to extract features as keys and values from a content glyph set and a style glyph set, respectively. Secondly, a query vector generated from a single glyph sample by the query encoder is applied to draw out proper features from the content and style (key, value) pairs via glyph-attention modules. Next, a decoder is used to recover a glyph from the queried features. Lastly, Adversarial lossesGoodfellow et al. (2014) with multi-task glyph discriminator are employed to stablize the training process. Experimental results demonstrate that our method is able to create robust results with superior fidelity. Less number of samples are needed and better performance is achieved when compared to the other state-of-the-art few-shot font generation methods, without utilizing supervision on locality such as component, skeleton, or strokes, etc.

## 1 Introduction

With the increasing popularity of Chinese characters globally, the demand for Chinese fonts is rising. However, it's very time consuming and expensive to design such font library. As we know, the number of letters in English alphabet is 52 (upper and lower cases), much smaller than that of the Chinese characters which is over 20,000. It takes about 6 months for an experienced designer to design a Chinese font library. Typically, the designer starts the font design with manually designing a few hundred of Chinese characters which encompass most of the radicals (i.e., the components of the characters). The rest of the characters will be completed by mixing and matching the radicals based on word vocabulary. However, it is very tedious and time consuming in both the initial design phase and the block building process. If only a few Chinese glyphs(e.g. 4 or 8) are needed by manual design and there is a tool that can automatically complete the font library of the 20,000+ characters, designers' efficiency will be greatly improved.

Recently, image to image translation based methods Jiang et al. (2019) have been proposed to generate fonts automatically. Most of these approaches are based on generative adversarial network Goodfellow et al. (2014), which was shown to be able to transfer one font style to another successfully. However, they need large scale paired or un-paired source and target domain dataset to train a model. It's inefficient because for every new font style, the pretrained model needs to be finetuned. Moreover, it is difficult and expensive to collect target font dataset such as Chinese historical calligraphic works, whose font glyphs may just have a small percentage left. But these methods typically require hundreds of font glyphs for training as a minimum.

In order to address the limitation of lack of dataset, several few-shot font generation methods have been developed Gao et al. (2019)Park et al. (2021), which don't demand a lot of glyphs

to train the model. They are capable of producing a complete font library with just few samples. Although these methods can fuse source content and target style representations successfully with few font glyphs unseen style during training, they tend to output characters with either wrong content or weak style.

The unsatisfactory results of recent few-shot font generation methods make us re-examine the style and content relationship in the glyphs. A Chinese glyph can be divided into two parts (i.e. content and style). From the content perspective, we recognize a glyph's meaning at a glance. It implies that human understands font's content through vision based on font global information. Inspired by this observation, we propose a content glyph-attention module to capture the global features from the content set of glyphs. On the other side, special stroke and radical details of a glyph determine the its style. In other words, glyph style is closely related to local information. Based on this hypothesis, we propose a style glyph-attention module to encapsulate pattern information from the style set of glyphs.

Our contributions are summarized as follows:

- We propose a generic few-shot font generation framework that bridges the gap between font content and style.
- We propose a content and a style glyph-attention modules to characterize overall structure information from the content set and local stroke details from the style set, respectively.
- Experiments demonstrate superior performance in terms of both quantitative and qualitative results compared with other cutting-edge few-shot glyph generation methods.

## 2 Related Work

### 2.1 Style Transfer

Few-shot Chinese font generation aims to combine the content and reference style. It is a special style transfer problem in essence. However, font design requires high accuracy in content structure, the result of traditional style transfer model for font generation is usually unsatisfactory. Gatys et al. (2016) proposed neural style transfer in VGG Simonyan & Zisserman (2014) feature space by reconstructing the gram matrix but with limited inference speed. Johnson et al. (2016) trained a fast neural style transfer(FNST) model, which is able to merge style and content representations in real-time. Although speed-up is achieved in FNST, a new style image is necessary to retrain the model for transfer. In order to solve this problem, Dumoulin et al. (2016) proposed a n-style transfer method which can embed n style features by using conditional instance normalization. Further improvement was made by Huang & Belongie (2017) by inventing AdaIN that can achieve arbitrary style transfer in real time without retraining a model.

### 2.2 Image to Image Translation

Although style transfer method has advantage in combining content and style representations, it may damage the content structure. For font generation, it is crucial to keep character semantics so that they are visually distinguishable and recognizable. Image to image translation method (pix2pix) Isola et al. (2017) was devised to transfer from the source domain to the target domain (e.g., from a sketch to a real cat drawing). Pix2pix differs from style transfer based methods in that it prevents the structure from being destroyed. Nevertheless, the result quality of pix2pix is heavily dependent on paired dataset that is expensive to collect. In order to tackle this problem, Zhu et al. (2017) moved a step further by introducing unpaired image to image translation method(CycleGAN), which maps the source domain to the target domain by using cyclic loss. Although CycleGAN made break-through by eliminating the necessity of paired dataset, it's inefficient because different (source, target) domain tuples require different models for inference. Huang et al. (2018) extended CycleGAN by implementing multi-domin transfer in a single model. Recently, Liu et al. (2019) put forward few-shot image to image translation using AdaIN.

## 2.3 Attention Models

Lately, attention mechanisms are widely adopted in several models in order to capture global dependencies Xu et al. (2015). In particular, self-attention (also known as intra-attention) calculates the response at a position in a sequence by attending to all positions within the same sequence Parikh et al. (2016). (SAGAN) Zhang et al. (2019) demonstrated that self-attention could improve image generation quality of the GAN model. (non-local) Wang et al. (2018) used a reformalized self-attention module in spatial-temporal domain between video sequences. To our knowledge, attention module has not yet been explored in font generation tasks. In this study, we apply content and style glyph attention modules to efficiently capture the characteristics of the font content and style sets.

## 2.4 Few-shot Font Generation Methods

The goal of few-shot font generation method is to make the generated glyphs akin to the style references, which are only a few samples, without re-training a new model. At the same time, the generated glyphs' semantic is unchanged. Recently, Zhang et al. (2018) proposed EMD model that leverages a pair of encoders to merge content and style. Gao et al. (2019) invented AGIS-Net to reach the few-shot font generation goal by transferring both shape and texture with a few reference samples. Unlike other methods, MX-Font Park et al. (2021) extracts multiple style features not explicitly conditioned on component labels, but automatically by multiple experts to represent different local concepts.

## 3 METHOD

As mentioned above, our goal is to generate stylized glyph images from a small number of reference samples. We design three encoders, two glyph attention modules and one decoder to form a Glyph-Attention Network(GANet). Details of the model architecture and loss functions are discussed in Sections 3.1 and 3.2.

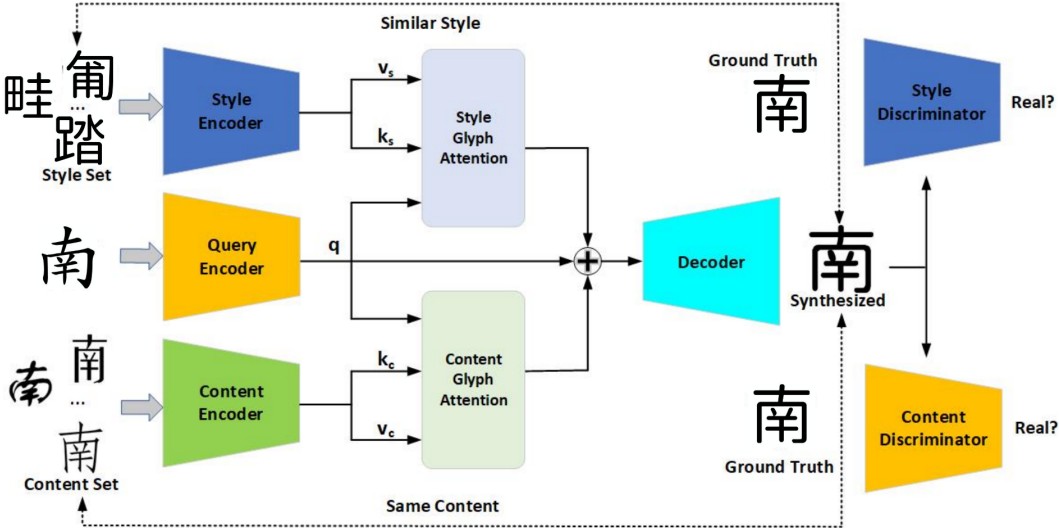

Figure 1: The framework of proposed few-shot font generation model GANet. A query encoder is used to extract glyph content from the content encoder and glyph style from the style encoder by two glyph-attention modules. They are subsequently added element-wise before being fed into the decoder. Two discriminators are employed to stablize the training process. For more details, please refer to the appendices.

### 3.1 Network architecture

As shown in Figure 1, GANet consists of one query encoder, one style encoder, one content encoder, two glyph-attention modules and one decoder. In order to stablize the training process and improve the synthesized results quality, two discriminators are adopted.

#### 3.1.1 Query, Content and Style Encoders

We formulate the content, style and query encoding process as trainable high-dimensional glyph feature extractors. The content encoder $E_c$ aims to extract semantic feature. The input of the content encoder is $X_c = \{x_1, x_2, ..., x_N\}$, each of them has the same content but different style, the output of the content encoder is $F_c = E_c(X_c) \in \mathbb{R}^{N \times H \times W \times C}$. Similarly, the style encoder $E_s$ is used to characterize style features from the the style set $X_s = \{y_1, y_2, ..., y_N\}$, which contains $N$ glyphs with the same style but different content. The output of the style encoder is $F_s = E_s(X_s) \in \mathbb{R}^{N \times H \times W \times C}$. The query encoder $E_q$ is devised to obtain a query feature vector $F_q$ that is essential for the glyph-attention modules to identify local style features from the style feature set $F_s$ and generate the most proper global content feature from the content feature set $F_c$. The input of the query encoder is a glyph image $X_q$ whose content is the same as the content in the content glyph set but with a new style. Output of the query encoder is $F_q = E_q(X_q) \in \mathbb{R}^{H \times W \times C}$. The three encoders $E_s$, $E_q$ and $E_c$ have identical architecture but they do not share weights. Symbols $H$, $W$ and $C$ represent feature map's height, width and channel respectively.

#### 3.1.2 Decoder

The style glyph-attention module is intended to query a local style feature tensor from the style feature set. Similarly, the objective of the content glyph-attention module is to acquire the most proper glyph global content from the content feature set. The outputs of the content, style, and query encoders are added element-wise before being fed into the decoder. As defined in (1)

$$F = SGA(q, k_s, v_s) + CGA(q, k_c, v_c) + F_q \tag{1}$$

$$O = Decoder(F) \tag{2}$$

Where $SGA$ and $CGA$ are abbreviations of Style and Content Glyph-Attention respectively, $k_s$ and $v_s$ are equal to $F_s$ which is from the style encoder. Similar to $SGA$, $k_c$ and $v_c$ are equal to $F_c$ which is from the content encoder. $q$ is equal to $F_q$ which is from the query encoder. $O$ is the synthesized result generated by the decoder.

#### 3.1.3 Multi-task discriminators

It has been demonstrated that generative adversarial networks(GAN) Goodfellow et al. (2014) are able to generate output with distribution close to that of the actual data. Therefore, we can leverage GAN to develop a model that outputs synthesized glyph images whose content and style comply the distributions of the input content and reference style. In principle, a conventional discriminator, which can discern the fake and real data, can perform the discrimination task independently to complete the generation task. However, the synthesized results are not satisfactory because they are often wrong in content or weak in style. To enhance the model performance, multi-task discriminators are proposed to further disentangle content and style information. They are called content $D_c$ and style $D_s$ discriminators respectively, based on SNGAN Miyato & Koyama (2018), which uses a trainable embedding matrix to obtain conditional information. $D_c$ allows closer content distributions between the model output and the real data, and $D_s$ helps the convergence of the output style to that of the real data.

### 3.2 Glyph attention

Most few-shot image generation methodsLiu et al. (2019) use adaptive instance normalization (AdaIn) Huang & Belongie (2017) to transfer image style such as texture and color to another which provide content like pose and structure. However, it's challenging to transfer

font style to another due to the fact that font does not have texture and color, just consists of some white and black strokes. Different from AdaIn, we propose style and content glyph-attention modules to extract style and content features from custom glyph sets. The proposed method Glyph-Attention Network (GANet) is named after the glyph-attention modules, which are shown in Figure 2.

### 3.2.1 Style glyph-attention

As we all know, font style mainly depend on local stroke details, so it means that local features of a glyph dictate styles. Based on this hypothesis, we propose the style glyph-attention module. The style encoder provides us a style feature set $F_s \in \mathbb{R}^{N \times H \times W \times C}$ which contains the desired font style. A style glyph-attention module is utilized to query the style feature from the style set. $v_s$ and $k_s$ denote the value and key in the attention module. They are equal to $F_s$. The query vector $q \in \mathbb{R}^{H \times W \times C}$ is derived from the query encoder. Key $k_s$ and query $q$ are first transformed into two feature spaces $f_s$, $g_s$, where $f_s(q) = W_{f_s}q$, $g_s(k_s) = W_{g_s}k_s$. We then reshape $g_s(k_s) \in \mathbb{R}^{NHW \times C}$ and transpose and reshape $f_s(q) \in \mathbb{R}^{C \times HW}$. Finally, they are used to calculate the attention map $\beta$, as defined in (3),

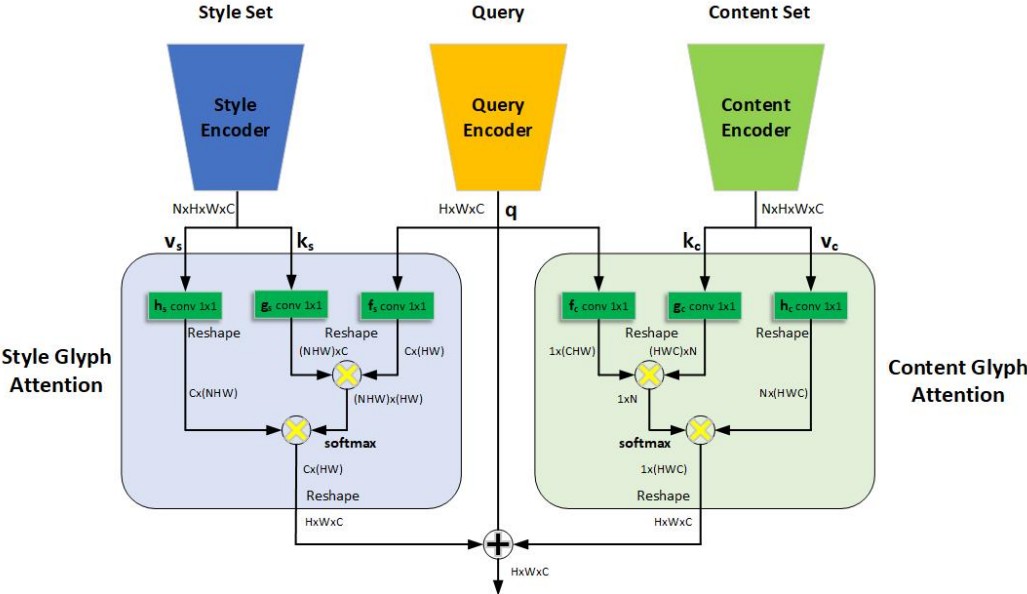

Figure 2: Glyph attention and content attention

$$\beta_{ij} = \frac{exp(\xi_{ij})}{\sum_{m=1}^{NHW} exp(\xi_{mj})}, \quad \xi = g_s(k_s)f_s(q) \quad \xi \in \mathbb{R}^{NHW \times HW} \tag{3}$$

The attention map $\beta_{ij}$ indicates the extent to which the model attends to the $i^{th}$ style location when synthesizing $j^{th}$. Where H, W, C are the height, width and channel of features from previous $1 \times 1$ conv layer. $N$ is the number of glyphs in the style set. The output of the style glyph-attention layer is $O_s = (O_1, O_2, ..., O_j, ..., O_{HW})$, $O_j \in \mathbb{R}^{1 \times C}$, where,

$$O_{ij} = \sum_{m=1}^{NHW} \tau_{im}\beta_{mj}, \quad \tau = h_s(v_s) \quad \tau \in \mathbb{R}^{C \times NHW} \tag{4}$$

where $\tau$ is transposed and reshaped from $h_s(v_s) = W_{h_s}v_s$. In addition, $W_{h_s} \in \mathbb{R}^{C \times C}$, $W_{f_s} \in \mathbb{R}^{C \times C}$ and $W_{g_s} \in \mathbb{R}^{C \times C}$ are the learned weights in the $1 \times 1$ conv layer of the style glyph-attention module. The final output is reshaped to $O_s \in \mathbb{R}^{H \times W \times C}$.

### 3.2.2 Content glyph-attention

Different from the style glyph-attention module, which queries the style features from the local spatial area, the content glyph-attention mainly queries content features globally. It is based on an assumption that content information resides in the overall structure of the skeletons. We define a content feature set $F_c \in \mathbb{R}^{N \times H \times W \times C}$ that is extracted by the content encoder from the content set $X_c = \{x_1, x_2, ..., x_N\}$. $v_c$ and $k_c$ denote the value and key in the attention module, which are equal to $F_c$. The query vector $q \in \mathbb{R}^{H \times W \times C}$ is identical to that in the style glyph-attention module. Similarly, key $k_c$ and query $q$ are transformed into two feature spaces $f_c$, $g_c$, where $f_c(q) = W_{f_c}q$, $g_c(k_c) = W_{g_c}k_c$. $g_c(k_c) \in \mathbb{R}^{N \times HWC}$ and $f_c(q) \in \mathbb{R}^{HWC \times 1}$ are then reshaped to calculate the attention map $\gamma$. As defined in (5),

$$\gamma_i = \frac{exp(\xi_i)}{\sum_{m=1}^{N} exp(\xi_m)}, \quad \xi = g_c(k_c)f_c(q) \quad \xi \in \mathbb{R}^{N \times 1} \tag{5}$$

The attention map $\gamma_i$ indicates the extent to which the model attends to the $i^{th}$ global content set location. H, W, and C are the height, width and channel of features from previous $1 \times 1$ conv layer. N is the number of glyphs in the content set. The output of the content glyph-attention layer is $O_c = (O_1, O_2, ..., O_j, ..., O_{HWC})$, $O_j \in \mathbb{R}$,

$$O_j = \sum_{m=1}^{N} \tau_{mj}\gamma_m, \quad \tau = h_c(v_c) \quad \tau \in \mathbb{R}^{N \times HWC} \tag{6}$$

where $\tau$ is reshaped from $h_c(v_c) = W_{h_c}v_c$. In addition, $W_{h_c} \in \mathbb{R}^{C \times C}$, $W_{f_c} \in \mathbb{R}^{C \times C}$ and $W_{g_c} \in \mathbb{R}^{C \times C}$ are the learned weights in the $1 \times 1$ conv layer of the content glyph-attention module. The final output is reshaped to $O_c \in \mathbb{R}^{H \times W \times C}$.

### 3.3 Loss function

Three individual loss functions are combined during model training: identity loss, feature matching loss, and multi-task adversarial loss.

### 3.3.1 Identity loss

In order to make the output distribution of the model agreeing to that of the target in the pixel and feature space, we employ the identity loss. As we know, perceptual loss Johnson et al. (2016) is an effective way to measure the distance between images in VGG feature space. Compared to pixel-wise loss functions such as Mean-Squared Error(MSE), perceptual loss can recover more high-frequency details. However, every coin has two sides, due to the downsampling operation (max-pooling) in VGG19, the perceptual loss function will result in partial information loss during reconstruction, especially at low-frequencies. To avoid losing low-frequency details, we add L1 loss as a complementary term with perceptual loss, as defined in (7),

$$L_{id} = \left\| \Phi_{3\_1}(T) - \Phi_{3\_1}(G(X_c, X_q, X_s)) \right\|_1 + \left\| T - G(X_c, X_q, X_s) \right\|_1 \tag{7}$$

Where $\Phi_{3\_1}$ is the feature map of VGG19 in relu3\_1. $X_c$ and $X_s$ are input content set and reference style set respectively, $X_q$ is the query image whose content is the same as the content set. $T$ is the target image whose content is the same as $X_c$, with style similar to $X_s$. $G$ is the generator that consists of a content encoder, a query encoder, a style encoder, a decoder and two glyph-attention modules as shown in Figure 1.

### 3.3.2 Feature matching loss

To stabilize the adversarial training, feature matching (FM) loss Salimans et al. (2016) is also applied in this study. Similar to the perceptual similarity measure, the FM loss uses discriminator network as the feature extractor, which contain a large amount of information about the glyph images. To obtain more disentangled representations in content and style, we use $D_c$ and $D_s$ to extract feature maps, as defined in (8),

$$L_{fm} = \sum_{i=1}^{l} \left\| D_c^{(i)}(T) - D_c^{(i)}(G(X_c, X_q, X_s)) \right\|_1 + \sum_{i=1}^{l} \left\| D_s^{(i)}(T) - D_s^{(i)}(G(X_c, X_q, X_s)) \right\|_1 \tag{8}$$

Where $D^{(i)}$ is the feature map of layer $i$ in the discriminator. We use all layers except fully-connected layer to extract features.

### 3.3.3 Multi-task adversarial loss

In the proposed framework, two condition discriminators $D_c$ and $D_s$ are used to improve model performance, each of which has its own mission. $D_c$ and $D_s$ aim at calibrating the model to attain accurate content and stronger style respectively. We utilize hinge version loss Lim & Ye (2017) as the adversarial objective function, as defined in (9-10),

$$L_G = -D_s(G(X_c, X_q, X_s)) - D_c(G(X_c, X_q, X_s)) \tag{9}$$

$$\begin{aligned} L_D =& max(0, 1 + D_c(G(X_c, X_q, X_s))) + max(0, 1 - D_c(T)) + \\ & max(0, 1 + D_s(G(X_c, X_q, X_s))) + max(0, 1 - D_s(T)) \end{aligned} \tag{10}$$

### 3.3.4 Total loss

A simple additive form is used for the total loss, as illustrated in (11-12),

$$L_\theta(G) = \lambda_1 L_{id} + \lambda_2 L_{fm} + L_G \tag{11}$$

$$L_\omega(D) = L_D \tag{12}$$

Where $\lambda_1$ and $\lambda_2$ are the hyper parameters of the generator loss function, $\theta$ and $\omega$ are the weights for the generator and discriminators respectively.

## 4 Experiments

### 4.1 Experimental settings

We use the Adam optimizer Kingma & Ba (2014) to optimize all models with hyperparameters $\beta_1 = 0$ and $\beta_2 = 0.9$. Using the two-time scale learning rates Heusel et al. (2017) strategy, we set the learning rate of all discriminators to $4e-4$ and the generator to $1e-4$. The number of both $X_c$ and $X_s$ is set to 8 and the size of all images is $128 \times 128 \times 3$. The weights in the loss function are chosen to be $\lambda_1 = 1$ and $\lambda_2 = 10$. We train the model with batch size of 4 for 500,000 iterations in a Nvidia GTX 1080Ti GPU with 12G memory.

Figure 3: Comparison of the results in 6 different font styles for each method

Table 1: Intersection Over Union (IOU) for many-shot and few-shot font generation methods

| Intersection Over Union | | | | | | | | |
|---|---|---|---|---|---|---|---|---|
| Type | Method | Style | | | | | | |
| | | 1 | 2 | 3 | 4 | 5 | 6 | Mean |
| many shot | zi2zi | 0.529 | 0.534 | 0.599 | 0.695 | 0.515 | 0.423 | 0.549 |
| | pix2pix | 0.590 | 0.652 | 0.695 | 0.807 | 0.616 | 0.537 | 0.650 |
| | CycleGAN | 0.206 | 0.272 | 0.262 | 0.465 | 0.325 | 0.312 | 0.307 |
| | ZiGAN | 0.600 | 0.664 | 0.702 | **0.812** | **0.623** | **0.550** | **0.659** |
| few shot | FUNIT | 0.355 | 0.448 | 0.327 | 0.571 | 0.366 | 0.289 | 0.393 |
| | AGIS-Net | 0.387 | 0.369 | 0.418 | 0.736 | 0.483 | 0.388 | 0.464 |
| | MX-Font | 0.260 | 0.362 | 0.382 | 0.648 | 0.490 | 0.295 | 0.406 |
| | GANet | **0.711** | **0.683** | **0.749** | 0.778 | 0.481 | 0.401 | 0.634 |

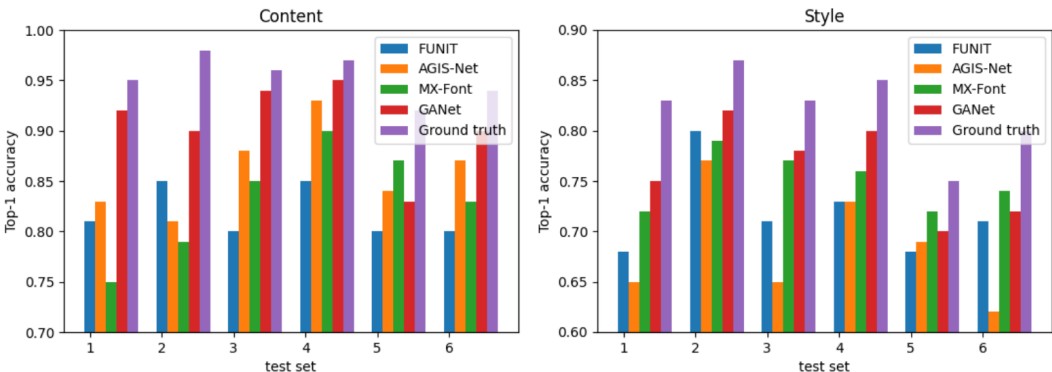

Figure 4: Top-1 accuracy of content and style in 6 different font styles for 4 few-shot font generation methods

## 4.2 Datasets and evaluation metrics

We collected 411 Chinese fonts conforming to the GB2312 standard. Each font contains 6,763 Chinese characters. We randomly selected 405 fonts as the training set and the remaining 6 fonts as the test set. Various metrics are used to evaluate the quality of the synthesized glyphs. To measure the similarity between the generated image and the target, intersection over union(IOU) is utilized. We further employ two classifiers with Inception-v3 Szegedy et al. (2016) backbone to distinguish between content and style labels in the test set.

## 4.3 Benchmarking

We compare our method with four many-shot font generation methods (i.e., zi2zi Tian, pix2pix Isola et al. (2017), CycleGAN Zhu et al. (2017), ZiGAN Wen et al. (2021)) and three state-of-the-art few-shot font generation methods (i.e., FUNIT Liu et al. (2019), MX-Font Park et al. (2021) and AGIS-Net Gao et al. (2019)). The four many-shot methods demand a lot of training data and a different model is required for each font style during inference. As such, for many-shot methods, we further split each test font library into 5763 glyphs for training and 1000 glyphs for testing. Therefore, each of the four many-shot methods generates 6 models. In comparison, few-shot methods including the proposed method GANet need only a few glyphs to transfer font style during inference and no re-training is necessary for different font styles. Different from many-shot method, the few-shot models are trained using the selected 405 font libraries. During inference, few-shot methods take a few glyphs (i.e. 1, 2, 4, 8) as references to synthesize the font library rather than training a new model. To make a fair comparison, we use the 1000 glyphs split from the test set to evaluate all models.

### 4.4 Quantitative evaluation

We assess the quality of the generated images in two aspects. Firstly, Intersection Over Union (IOU) is used to measure the similarity between the synthesized and the ground truth. Higher IOU score indicates better result. As shown in Table 1, we randomly sample four reference images from the 5763 glyphs of one test font, based on which each model generates 1000 glyphs. There are a total number of 6 fonts in the test dataset. The metrics in the table are calculated based on the $1000 \times 6$ glyphs for each model and averaged. Table 1 shows that our method outperforms previous state-of-the-art few-shot approaches by a wide margin and generate comparable results to the many-shot approaches. Howerver, our method needs only few reference samples and one-time training. On the contrary, many-shot methods demand re-training using thousands of glyphs for each new font style.

IOU can evaluate the similarity in pixel-level, but cannot represent the style and content in essence. In addition, pixel-based metrics are often inconsistent with human visual perception. To improve the reliability of the evaluation, we trained two classifiers to identify the content and style classifications respectively and then use them to evaluate model prediction accuracies for the test set. As shown in Figure 4, MX-Font and FUNIT show comparable performance in terms of style accuracy, but they show lower performance in content preservation. AGIS-Net shows comparable performance in content accuracy, but weak in style. In other words, FUNIT and MX-Font focus only on styling and fails to preserve content structure. AGIS-Net focuses on content and fails to transfer more style.

### 4.5 Qualitative analysis

To demonstrate the model's performance visually, we evaluate the models using the remaining 6 fonts in the test set as reference glyphs. Figure 3 presents the many-shot and few-shot methods generated images from the test dataset. It is observed that many-shot methods except ZiGAN always fail in content preservation as shown in the red box. Among few-shot methods, FUNIT learns the style but the local structure in the generated glyphs is weak as highlighted by the yellow box. Compared with FUNIT, AGIS-Net has better performance in content preservation. However, it has poor performance in capturing global and local styles (black box). MX-Font also succeeds in learning the style of the reference, but the content structure is unsatisfactory (blue box). Overall, our proposed method generated best results qualitatively compared to other methods both in terms of content preservation and style transfer.

## 5 Conclusion

In this paper, we propose a glyph-attention module (GANet) for few-shot font generation. GANet requires only a few reference glyphs as input to generate high-quality glyph images of the same style for any Chinese character. The model includes three trainable encoders to extract query, content and style features, and two glyph-attention modules to query global content and local style features, and a decoder for merging queried features to obtain stylized glyph images. To achieve better performance, multi-task discriminators are employed to minimize the distribution difference between the generated results and the ground truth. Quantitatively and qualitatively, the experimental results show that our model accomplishes the state-of-the-art compared with other cutting-edge few-shot glyph generation methods.

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

## A  Appendix

### A.1  Network architecture

| $x \in R^{128 \times 128 \times 3}$ |
| --- |
| ResBlock down 64 |
| ResBlock down 128 |
| ResBlock down 256 |
| ResBlock down 512 |
| ResBlock 512 |
| ResBlock 512 |

(a) Encoder

| $x \in R^{8 \times 8 \times 512}$ |
| --- |
| ResBlock up 512 |
| ResBlock 512 |
| ResBlock up 256 |
| ResBlock 256 |
| ResBlock up 128 |
| ResBlock 128 |
| ResBlock up 64 |
| ResBlock 64 |
| BN, ReLU, Conv 3, Tanh |

(b) Decoder

| $x \in R^{128 \times 128 \times 3}, y \in R$ |
| --- |
| ResBlock down 64 |
| ResBlock down 128 |
| ResBlock down 256 |
| ResBlock down 512 |
| ResBlock down 512 |
| ResBlock down 512 |
| Global sum pooling |
| Relu |
| Inner product |
| Dense 1 |

(c) Discriminator

Tables (a), (b) and (c) present network architectures of the encoder, decoder and discriminator respectively. The proposed GANet consists of three encoders (i.e., query, content and style encoders). They have the same architecture but don't share the weights. The two discriminators (content and style) also have the same architecture and don't share the weights. The slight difference between them lies in the inner product module Miyato & Koyama (2018), due to the difference in the count of labels in content and style. The decoder's input is a feature map that is from the three encoders as defined in 1. And all three network architectures consist of ResBlcok Miyato & Koyama (2018).

### A.2  Ablation study

#### A.2.1  Does glyph-attention really work?

In order to verify the effectiveness of content and style glyph-attention modules, we control the feature flows before being fed into the decoder as defined in 1. As shown in Figure 5, firstly, we only feed the query feature to the decoder (the first row) and the synthesized glyphs fail to capture content because the query features carry little content information. Secondly, when we switch on the query and content feature flows at the same time (the

second row), the synthesized glyphs bear correct content without correct style. When we switch on the query and style flows, there is font style but no content information as input. The hypothesis is that the content glyph-attention module mainly focuses on the global features of the content set, so $q + c$ can synthesize correct content. On the other side, the style glyph-attention module query the local style features from the style set. As shown in Fighre 5, $q + s$ shows chaotic results in many local style patches. Finally, when we switch on query, content and style feature flows, the synthesized results contain both correct content and style.

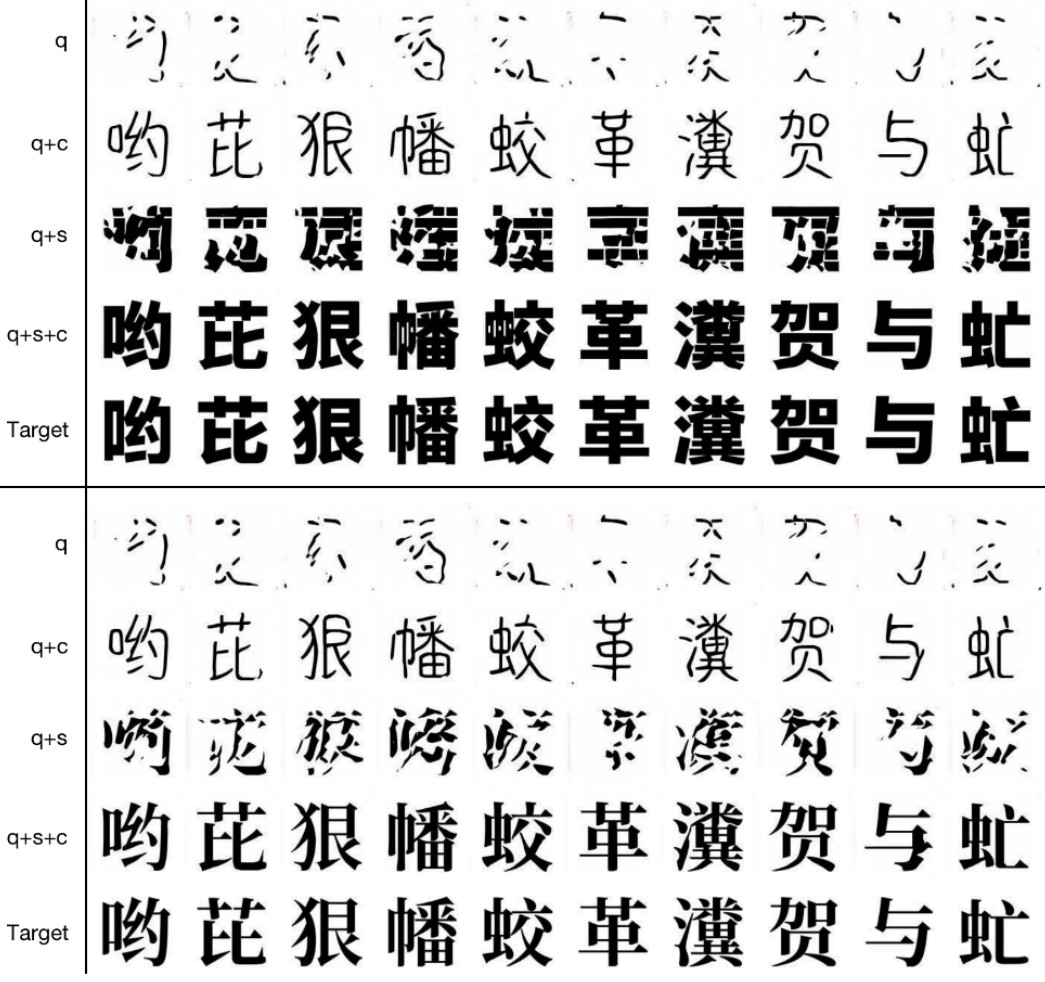

Figure 5: The function of glyph-attention. The first column q, c and s are noted as query, content and style feature flow respectively

### A.2.2 Relationship between shot and content-style

Because glyph-attention modules query content and style features from the content and style sets, the length of which impact the performance of the model. In order to figure out the relationship between shot number and model performance, we use two classifiers to identify the synthesized glyphs from the content and style aspects respectively. On the left side in Figure 6, we fix the length of the style set to eight and vary the shot number of the content set. It is found that when the shot number changes from 1 to 2, the content top-1 accuracy improves significantly. However, the performance gain is limited when the shot number increases from 2 to 16. Similar observation is found in the right side of the figure.

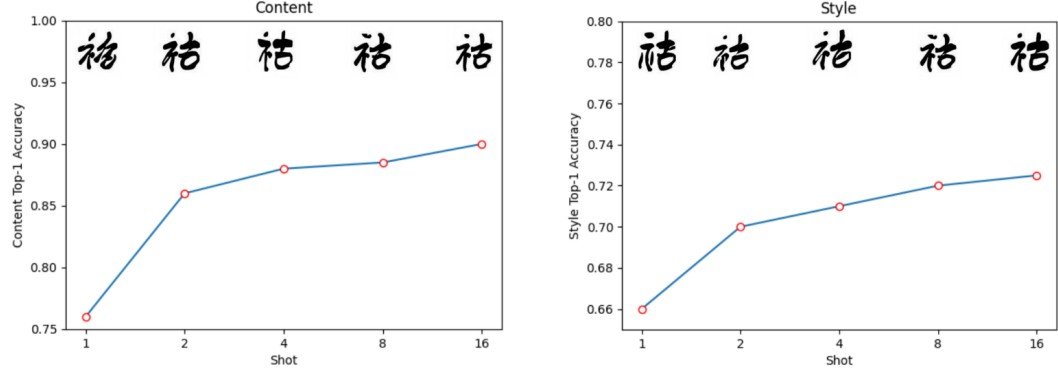

Figure 6: Top-1 accuracy of content and style. On the left figure, fix the count of style set to 8, and change the shot number of content set; on the right figure, fix the count of content set to 8, and change the shot number of style set.

### A.2.3   Interpolation in content and style

To show that our content and style representations are semantically meaningful, we provide the content and style interpolation results in Figure 7. In the content interpolation, we can observe that the style is maintained when interpolated from one glyph to another. Whereas in the style interpolation, the content structure is well preserved. It is concluded that our model is able to disentangle the content and style representations well.

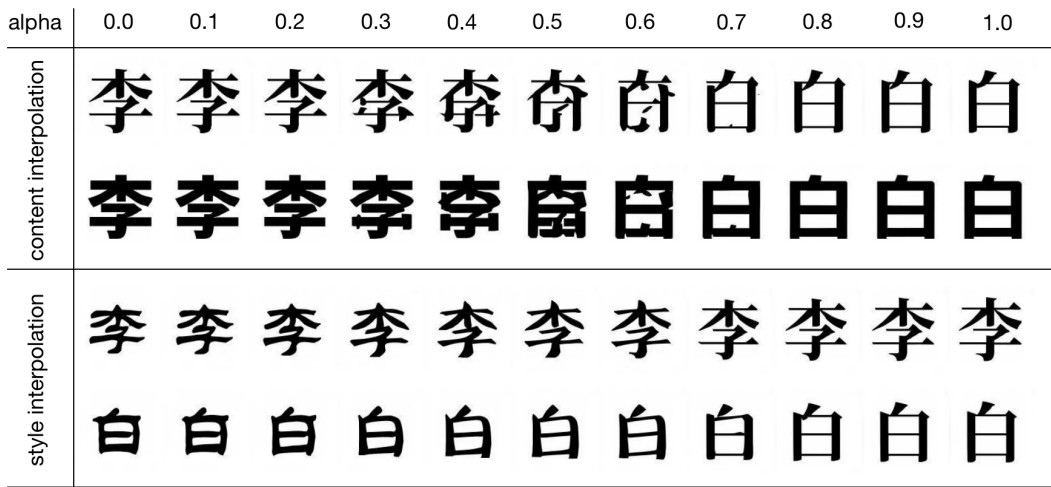

Figure 7: Interpolation in content and style. For thr content interpolation, because the query also contain the content information, we use $q+c$ feature that is from content glyph-attention module and query encoder as the interpolation content feature. For the style interpolation, we just use the $s$ feature that is from style glyph-attention module to interpolate.

### A.3   More synthesised results

Figure 8: More synthesised results. Style set and content set are both contain 4 glyphs to synthesis new font.

Figure 9: More synthesised results. Style set and content set are both contain 4 glyphs to synthesis new font.

