# OpenReview forum: "GANet: Glyph-Attention Network for Few-Shot Font Generation"
_ICLR.cc/2022/Conference — ICLR 2022 Submitted_

### Official Review · Reviewer_6xeo · 2021-11-01

**Correctness:** 3
**Technical Novelty And Significance:** 3
**Empirical Novelty And Significance:** 3
**Recommendation:** 5
**Confidence:** 5

**Main Review:**


[Paper strengths]
The motivation is clear and reasonable. Different from image style transfer which usually adopts Ada-IN to transfer global and statistic information (e.g., texture and color), the paper proposes style and content glyph-attention modules to extract style and content features from custom glyph sets.

[Paper weakness]
Experiment:
1. The experiment setting and comparison need to be clarified:
  - In Section 4.1, the paper claims that the proposed methods use 8 style reference images and 8 content reference images. In contrast, in Fig 4, the number of styles and content reference images are used for the compared methods and the proposed model should be explained.
  - Which fonts does the model use as query images and content reference images?
2. More evaluation metrics for image generation should be used, such as FID、LPIPS.
3. Lacks comparison to latest font generation methods:
   [1] Few-shot Font Generation with Localized Style Representations and Factorization. S Park, S Chun, J Cha, B Lee, H Shim. AAAI 2021.
   [2] DG-Font: Deformable Generative Networks for Unsupervised Font Generation.
Y Xie, X Chen, L Sun, Y Lu. CVPR 2021.
   [3] RD-GAN: Few/Zero-Shot Chinese Character Style Transfer via Radical Decomposition and Rendering. Y Huang, M He, L Jin, and Y Wang. ECCV 2020.

Modeling:
4. The query image is one of the content images. What attention does content glyph attention learn? I would like to see the visualization of the attention map for both style glyph attention and content glyph attention.
5. The writing is not clear. In Sec 3.3.3, more explanation and notation should be clarified. Also, I would like to know how many binary classification heads are used for content and style discriminators.


**Summary Of The Paper:**

The paper proposes a glyph-attention network for few-shot font generation. They claim that fonts’ content features are basically global features and style features are related to local features. They propose a style glyph attention to capture the global features from the content set of glyphs, and a content glyph attention module to encapsulate pattern information from the style set of glyphs. The motivation is clear and reasonable but there are still some major issues in model construction and the experiment.

**Summary Of The Review:**

Overall, the paper’s motivation is clear and reasonable. However, there are still some issues on modeling and experiments that need to be clarified. Also, the overall writing is not clear.

---

### Official Review · Reviewer_oLEK · 2021-11-01

**Correctness:** 3
**Technical Novelty And Significance:** 2
**Empirical Novelty And Significance:** 2
**Recommendation:** 5
**Confidence:** 5

**Main Review:**

Detailed analyses on the strengths and weaknesses of the paper are listed as follows:

-- Strengths of the paper:
Few-shot font synthesis is a challenging and ongoing task in the areas of CV, CG, and AI. Existing approaches mainly rely on the disentanglement of glyph content and style. Typically, two encoders that aim to extract content and style features, respectively, are adopted, and content and style information are disentangled by applying various techniques and/or training strategies. The proposed GANet utilizes a different network architecture motivated by the attention mechanism. Such kind of network design is novel in solving the problem of font synthesis. The utilization of style glyph-attention and content glyph-attention seems reasonable for few-shot font generation. Both quantitative and qualitative experiments have been conducted to verify the effectiveness of the proposed method. To sum up, I think the key strength of this paper is to design a novel network inspired by the self-attention mechanism to handle the challenging task of few-shot font generation.

-- Weaknesses of the paper:
1) Some technical details of the proposed method are doubtful, where more explanations and experimental evidence are required. More specifically, the query branch plays an important role in the whole network but why the branch should be designed in the current form is unclear. Basically, since the content of the query glyph image is the same as glyph images in the content set, applying the self-attention operation seems enough to achieve the content glyph attention. Why should we use a glyph image with the same content in a different font style as the query for the content attention module? Similarly, many readers might also raise the question of why the glyph image with the target content in a different style can be used as the query for the style attention module? It seems that there is no relationship regarding the target style information between the query feature and the value (key) feature in the style glyph attention module. Furthermore, directly adding the outputs of the two attention modules with the query feature also seems less meaningful compared to the concatenation of these three features. Synthesis results of the proposed methods with these two settings should be compared. At last, the authors simply mentioned that the query glyph image should be with the same content as the content set but in a new style, will the selection of different query font styles affect the performance of GANet?

2) Experimental results presented in this paper are not convincing enough to support the authors’ statement that “Less number of samples are needed and better performance is achieved when compared to the other state-of-the-art few-shot font generation methods”. Firstly, during inference, only 8 glyph images in the target style are required to be fed into the style encoder as reference and no finetuning operation is needed. Therefore, the quality of synthesized glyph images might heavily rely on the training dataset. Namely, when the target font style is similar to some fonts in the training dataset or can be properly interpolated using those existing font styles, high-quality synthesis results will be obtained. Otherwise, few-shot generative models without fine-tuning will more likely perform unsatisfactorily. Therefore, it is important to show the nearest neighbors of synthesized glyph images in the training dataset to demonstrate that the model is still capable of handling unseen and special font styles. For example, the authors should construct the test dataset by selecting some fonts with quite special styles or asking professional designers to create some glyph samples in artistic font styles which are quite different from those in the training dataset. However, as we can see from the experiment section (see Fig. 3), the 6 fonts in the test dataset all possess commonly-seen font styles. Most probably, fonts with very similar styles as those 6 fonts can be found in the training dataset. At least, the NNs of the synthesized glyph images in the training dataset should also be shown in Fig. 3. What is worse, when we zoom in Fig. 3, it can be clearly observed that there exist many artifacts/distortions in the synthesis results of GANet, especially in the last two columns, qualitatively denying the authors’ conclusion that their proposed GANet outperforms other state-of-the-art few-shot font synthesis methods.

3) There exist many typos and informal expressions. The whole manuscript needs careful proofreading to improve the presentation quality. Here, I just list some examples of the writing problems and more can be found by carefully checking.
-1- The “\cite” commend should be used instead of “\shotcite”, which results in wired sentence like “Lastly, Adversarial lossesGoodfellow et al. (2014)…” in the abstract:
-2- In Section 3.2.1: “font style mainly depend on local stroke details,” ->?
-3- Below Eq. (7): ” Where Φ3_1 is …” ->”where Φ3_1 is …”
-4- In Section 2.3: “(SAGAN) Zhang et al. (2019) ……. (non-local) Wang et al. (2018) …” ->?
-5- In Page 5: “font does not have texture and color, just consists of some white and black strokes.” -> ?
and more…


**Summary Of The Paper:**

This paper proposed a few-shot font generation method, GANet. The key idea is to design glyph-attention modules including the style glyph-attention module and the content glyph-attention module to recover the glyph in the target style from the queried glyph. The multi-task adversarial loss was also employed to further improve the synthesizing performance. Experiments conducted on a dataset constructed by the authors verified the effectiveness of the proposed method in handling the task of few-shot Chinese font synthesis.

**Summary Of The Review:**

As analyzed in my above comments, I think the quality of this paper is slightly below the bar of ICLR. I will be happy to read the author's responses and discuss with other reviewers to make my final decision.

---

### Official Review · Reviewer_ihg3 · 2021-11-02

**Correctness:** 3
**Technical Novelty And Significance:** 2
**Empirical Novelty And Significance:** 2
**Recommendation:** 5
**Confidence:** 4

**Main Review:**

Strengths:
+ the paper is well written and easy to follow in most parts
+ the design of glyph attention component fits with problem well
+ experiment results are comprehensive

Weaknesses:
- I am confused why there is a query and a separate content set. They all have the same content, and there seems to be no requirement on their styles. Then why not treat them in the same way? If we have to have a single query, how to select the query?
- The method uses 400+ Chinese fonts as training data, and 6 as test set. It is very likely there are some training fonts very similar to the test ones. So there should be a very strong baseline where you can find the most similar training fonts to the query, and directly use the glyph in the similar training font with the same content as target. The authors should provide such baseline results to illustrate how novel the test font styles are compared with what's in the training set.

Minor issues:
- The citation format in many places is incorrect.
- Why both the style set and glyph set needs to have the same number of glyphs?
- The description "k_s and v_s are equal to F_s" in Sec 3.1.2 is somewhat confusing. I think this is basically a self attention model. The "key" and "value" are used more often in the context of a memory network.



**Summary Of The Paper:**

This paper considers Chinese glyph font style transfer problem with few reference inputs. The model has three encoders for query, style and content references; one decoder for target generation; and two discriminators for style and content. The main novelty comes with the glyph attention design with both local and global structure for style and content.

**Summary Of The Review:**

I like the design of the glyph attention part of the model. However, I am concerned about the use of a query and a content set with unclear relationship between them. Also, seeing the baseline comparison proposed above will give more evidence about how challenging is the problem and how well the method performs.

---

### Official Review · Reviewer_GLJu · 2021-11-05

**Correctness:** 4
**Technical Novelty And Significance:** 1
**Empirical Novelty And Significance:** 2
**Recommendation:** 3
**Confidence:** 4

**Main Review:**

Strengths:
1.	The proposed content and style glyph-attention modules are novel to some degree;
2.	The proposed framework outperforms other baseline methods quantitatively and qualitatively.

Weaknesses:
1.	Technical novelty is limited. The proposed framework is essentially a transformer variant. Although this work is probably the first to apply a Transformer-like model on few-shot font generation, there exist works that have attempted to apply in closely related tasks, like style transfer [1].
2.	Evaluation is insufficient. For quantitative comparison, only IoU and classification accuracy are provided. It will be more convincing to provide comparison results on FID and SSIM to show the effectiveness of the proposed framework.
3.	References and baseline methods are missing. [2] and [3] are both proposed for font generation but are not mentioned.
4.	Experimental details are not clear. (1) how to conduct the classification task on content, as mentioned in Section 4.4? 2. As mentioned in Section 3.3.3, the discriminators are conditional. What is the input of the discriminators?
5.	More insights are needed. It would be better to move the discussion in the Appendix to the main body of the manuscript.

References:
[1] StyleFormer: Real-time Arbitrary Style Transfer via Parametric Style Composition, ICCV 2021
[2] Multi-Content GAN for Few-Shot Font Style Transfer, CVPR 2018
[3] DG-Font: Deformable Generative Networks for Unsupervised Font Generation, CVPR 2021


**Summary Of The Paper:**

In this submission, the authors proposed a framework for few-shot font generation where the key components are content and style glyph-attention modules. They demonstrated the effectiveness of the proposed framework on a newly-collected dataset by comparing the IoU of different methods.

**Summary Of The Review:**

This paper proposed a Transformer-like model for few-shot font generation. However, its technical novelty is limited, evaluation is not sufficient, and the discussion is not thorough. The authors can improve this paper by including more recent works as baselines and comparing them using more evaluation metrics.

---

### Decision · Program_Chairs · 2022-01-20

**Decision:**

Reject

**Comment:**

This paper proposes a framework for few-shot font generation. Reviewers thought that the model was well-suited to the task. They were split on clarity, with some saying that the paper was easy to follow and others saying that it lacked sufficient detail. Overall reviewers found the technical novelty limited, saying that the approach was a “transformer variant”, while it was the first to apply a Transformer-like model on few-shot font generation, there wasn’t sufficient novelty in this task to have broad appeal to the ICLR community. The reviewers also pointed out some deficiency in the evaluation, concerning the chosen metrics (multiple reviewers requesting fidelity metrics) and missing baselines. Some reviewers posed questions to the authors but the authors did not respond to the reviews. There is a clear consensus to reject the paper.